# A Novel Flower-Like Ag/AgCl/BiOCOOH Ternary Heterojunction Photocatalyst: Facile Construction and Its Superior Photocatalytic Performance for the Removal of Toxic Pollutants

**DOI:** 10.3390/nano9111562

**Published:** 2019-11-04

**Authors:** Shijie Li, Bing Xue, Genying Wu, Yanping Liu, Huiqiu Zhang, Deyun Ma, Juncheng Zuo

**Affiliations:** 1Key Laboratory of key technical factors in Zhejiang seafood health hazards, Institute of Innovation & Application, Zhejiang Ocean University, Zhoushan 316022, China; xb1725621827@163.com (B.X.); zhanghuiqiu2006@163.com (H.Z.); 2College of Marine Science and Technology, Zhejiang Ocean University, Zhoushan 316022, China; zjuncheng@zjou.edu.cn; 3Longquan Branch of Lishui Municipal Ecological Environment Bureau, Longquan 323700, China; hannah29@163.com; 4School of Food and Pharmaceutical Engineering, Zhaoqing University, Zhaoqing 526061, China; mady@zqu.edu.cn

**Keywords:** Ag/AgCl/BiOCOOH, ternary heterojunction, toxic pollutants, photocatalysis

## Abstract

Novel 3D flower-like Ag/AgCl/BiOCOOH ternary heterojunction photocatalysts were fabricated by the solvothermal and in-situ precipitation methods, followed by light reduction treatment. The Ag/AgCl nanoparticles were homogeneously distributed on 3D BiOCOOH microspheres. These obtained catalysts were characterized by XRD, SEM, TEM, diffuse reflectance spectra (DRS), and photoluminescence (PL). As expected, they exhibited extraordinary photocatalytic capabilities for the elimination of rhodamine B (RhB) and ciprofloxacin (CIP) under simulated sunlight, the results revealed that the Ag/AgCl/BiOCH-3 with 20 wt.% of Ag/AgCl possessed the maximum activity, and the rate constant for the RhB degradation reached up to 0.1353 min^−1^, which was about 16.5 or 12.2 times that of bare BiOCOOH or Ag/AgCl. The PL characterization further verified that Ag/AgCl/BiOCOOH heterojunctions were endowed with the effective separation of photogenerated carriers. The excellent photocatalytic ability of Ag/AgCl/BiOCOOH could be credited to the synergistic interactions between Ag/AgCl and BiOCOOH, which not only substantially widened the light absorption, but also evidently hindered the charge recombination. The trapping experiments revealed that the dominant reactive species in RhB removal were h^+^, •OH, and •O_2_^−^ species. In addition, Ag/AgCl/BiOCOOH was quite stable and easily recyclable after multiple cycles. The above results imply that the 3D flower-like Ag/AgCl/BiOCOOH ternary heterojunction photocatalyst holds promising prospects in treating industrial wastewater.

## 1. Introduction

The massive development of global industry has brought about the aggravation of environmental pollution. Especially, contamination of refractory pollutants (e.g., industrial dyes and pharmaceutical antibiotics) in the aquatic environment poses an overwhelming threat to human health. To mitigate this problem, scientists have been devoted to exploring effective and eco-friendly techniques [1,2]. Among them, semiconductor-mediated photocatalysis has been deemed as a sustainable and efficient technology for environmental protection [3,4,5,6,7,8]. In the field of photocatalysis, BiOCOOH (Eg = ~3.7 eV) has recently emerged as a promising candidate for wastewater treatment by virtue of its unique layer architecture, high chemical stability, and non-toxicity [9,10]. However, the photocatalytic performance of BiOCOOH is still unsatisfactory due to the insufficient sunlight absorption and rapid carrier recombination. Therefore, great efforts have been made to reinforce the photocatalytic capability of BiOCOOH [9,11,12,13,14,15,16,17]. Of note, the fabrication of semiconductor heterojunctions is an effective route to conquer these drawbacks [11,12,13,14,15,17,18]. In this regard, searching for an appropriate candidate to be integrated with BiOCOOH for the efficient removal of pollutants has been very anticipated but challenging.

Up to now, Ag/AgCl has been widely utilized to modify different semiconductors for triggering extraordinary photocatalytic capability because of its ability to efficaciously improve the visible light (VL) response using the surface plasmon resonance (SPR) effect of metallic Ag and boost the separation of electrons and holes [19,20,21,22,23,24,25]. For instance, Ag/AgCl/titanium phosphate [25], Ag/AgCl/TiO_2_ [22], and Ag/AgCl/SrTiO3 [21] have been constructed and have displayed superior VL photocatalytic activity. Based on band theory, the band structures of Ag/AgCl and BiOCOOH are suitable for the generation of a novel plasmonic “type II” heterojunction, which could efficiently promote the separation of photo-excited charge carriers during the photocatalytic process. Enlightened by the above consideration, the novel hierarchical heterostructure of BiOCOOH decorated with Ag/AgCl nanoparticles can be expected to be a highly efficient photocatalyst for pollutant elimination under simulated sunlight.

Herein, in this work, a novel 3D flowerlike Ag/AgCl/BiOCOOH ternary heterojunction photocatalyst has been successfully synthesized through a facile solvothermal-precipitation-photoreduction route. Ag/AgCl NPs were evenly deposited in BiOCOOH micro-flowers, enabling fast separation of carriers and better absorption of light via the multi-reflection of light. The underlying properties and photocatalytic capability toward the removal of rhodamine B (RhB) and ciprofloxacin (CIP) in aqueous solution were systematically investigated. The possible photocatalysis mechanism regarding the transfer and segregation of charge carriers was proposed. This study offers deep insight into the design of highly efficient ternary heterojunction photocatalysts for wastewater treatment.

## 2. Materials and Methods 

### 2.1. Chemicals

All chemicals of analytical grade were purchased from Chinese Chemical Reagent factory (Shanghai, China) and utilized as was received without further purification.

### 2.2. Photocatalysts Preparation

**Synthesis of BiOCOOH microspheres**: First, 1 mmol of Bi(NO_3_)_3_·5H_2_O was dissolved in the solution of 25 mL of glycerol, 10 mL of H_2_O, and 10 mL of DMF with continuous stirring for 2 h. After that, the resulting solution was put into a 50 mL autoclave and then reacted at 160 °C for 24 h. When the reaction completed, BiOCOOH microspheres were rinsed repeatedly and then dried at 60 °C overnight.

**Synthesis of AgCl/BiOCOOH heterojunctions:** First, 0.32 g of BiOCOOH was evenly dispersed in 70 mL of solution containing 0.143 g of AgNO_3_ and 0.15 g of polyvinylpyrrolidone (PVP) under ultrasonic conditions to form solution A, and the resultant solution A was magnetically stirred for 1 h. Subsequently, 0.36 g of NaCl was dissolved in 20 mL of deionized water to form solution B. Then, solution B was injected into solution A by employing a syringe pump at a speed of 3 mL/h under vigorous agitation (1000 r/min), and the system was further stirred for 1 h. Afterwards, the 30 wt.% AgCl/BiOCOOH sample with a mass ratio of AgCl to BiOCOOH of 30% was totally rinsed to remove Na^+^ and Cl^−^ ions and then dried at 70 °C. 

**Synthesis of Ag/AgCl/BiOCOOH ternary heterojunctions:** 0.4 g AgCl/BiOCOOH was suspended in the solution of 30 mL H_2_O and 30 mL ethanol, followed by stirring for 30 min. After that, the mixture was irradiated by a 300 W Xe lamp for 2 h with continuous stirring. The as-prepared 30 wt.% Ag/AgCl/BiOCOOH was washed with deionized water thoroughly and dried at 70 °C in a vacuum oven and denoted as Ag/AgCl/BiOCH-4. Similarly, 5 wt.% Ag/AgCl/BiOCOOH (Ag/AgCl/BiOCH-1), 10 wt.% Ag/AgCl/BiOCOOH (Ag/AgCl/BiOCH-2), 20 wt.% Ag/AgCl/BiOCOOH (Ag/AgCl/BiOCH-3), AgCl, and Ag/AgCl were also synthesized. 

### 2.3. Characterization

The crystal constituents of the as-prepared samples were determined by using a X-ray diffraction (XRD, MiniFlex 600, Rigaku, Japan) diffractometer (Cu-Ka radiation at 40 kV and 15 mA, λ = 0.15418 nm, 2θ in the range from 20 to 80°). Microstructures of the as-fabricated catalysts were observed using a scanning electron microscope (SEM, Hitachi S-4800, Tokyo, Japan) and transmission electron microscope (TEM, Tecnai G2F20, Philips, Amsterdam, The Netherlands). UV-vis diffuse reflectance spectra (DRS) of the as-fabricated samples were collected using a UV-vis spectrophotometer (Shimadzu UV-2600, Tokyo, Japan). Photoluminescence (PL) analyses were implemented on a spectrophotometer (Hitachi F-7000, Tokyo, Japan). 

### 2.4. Photocatalytic Tests

Photocatalytic tests were conducted with RhB or CIP as the target pollutant over different samples under ambient conditions [26,27]. A 300 W Xe lamp (NeT HSX-300, Beijing, China) without a light filter to remove the UV light was employed as the simulated sunlight source. In each test, 30 mg of catalysts was first added into 80 mL of RhB (10 mg/L) or CIP (10 mg/L) aqua-solutions in a 250 mL beaker. Afterward, the resulting suspension was continuously agitated for 60 min in the dark. After that, the lamp was switched on to trigger the photocatalytic reaction. During the photocatalytic reaction, 3 mL of suspension was collected at a predetermined time interval and centrifuged (5000 rpm, 5 min) to separate aliquot from solid particles for further assessment of the photodegradation rate. The absorbance was monitored using an UV-vis spectrophotometer (Shimadzu UV-2600), and the normalized concentration variations of RhB and CIP were obtained based on their corresponding maximum absorption at 554 and 276 nm, respectively. The degradation rate (*η*) of RhB or CIP was estimated by the following equation: *η* = (*C*_0_ − *C_t_*)/*C*_0_ × 100%, where *C*_0_ and *C_t_* are the absorbance of the pollutant solution before irradiation and the absorbance of the pollutant solution at irradiation time *t*, respectively. 

The reusability and stability of Ag/AgCl/BiOCH-3 were measured via repeating the uniform experiment for six runs, and each run involved 30 min of dark adsorption and 30 min of reaction. After each run, the photocatalyst was first separated from the RhB aqueous solution via centrifugation (5000 rpm, 3 min), and then rinsed thoroughly with deionized water and ethanol, dried at 70 °C for 10 h, and subsequently subjected to the next run.

Mineralization degrees of RhB solution with the as-fabricated samples as the catalysts during the reaction were determined via measuring the total organic carbon (TOC) on a TOC analyzer (Shimadzu TOC–LCSH/CPH, Tokyo, Japan).

## 3. Results and Discussion

### 3.1. Structure and Morphology

The crystalline phases of BiOCOOH, AgCl, and Ag/AgCl/BiOCOOH heterojunctions with various contents of Ag/AgCl were characterized by XRD (Figure 1). For pure BiOCOOH, all the diffraction peaks coincided well with those of the tetragonal BiOCOOH (JCPDS 35-0939) [11,15]. Pristine AgCl was crystalized in a cubic phase (JCPDS 31-1238) with distinct diffraction peaks at 27.8°, 32.4°, 46.3°, 54.6°, 57.3°, and 74.5°, corresponding to the (111), (200), (220), (311), (222), and (331) crystal facets, respectively [23,28]. As to the Ag/AgCl/BiOCOOH heterojunctions, besides the peaks from BiOCOOH, several diffraction peaks belonging to Ag and AgCl were also detected, proving the successful fabrication of Ag/AgCl/BiOCOOH heterojunctions. Notably, as the content of the Ag/AgCl increased, the intensities of Ag/AgCl peaks gradually improved. Additionally, no other signals of impurities were detected, signifying the high purity of these heterojunctions. Meanwhile, the introduction of Ag/AgCl NPs did not lead to the phase change of BiOCOOH, implying that the Ag/AgCl NPs were probably coated on the BiOCOOH nanosheets instead of covalently incorporating into the crystal lattices of BiOCOOH.

The morphological and microstructural features of BiOCOOH and the Ag/AgCl/BiOCOOH heterojunction (Ag/AgCl/BiOCH-3) were investigated based on SEM and TEM measurements (Figure 2 and Figure 3). Pristine BiOCOOH possessed 3D hierarchical flower-like structures (diameter: ~2 to 3 μm) assembled by numerous 2D nanosheets (size: ~400 to 900 nm) (Figure 2a,b). When combined with the Ag/AgCl NPs, the as-prepared Ag/AgCl/BiOCOOH heterojunctions still consisted of hierarchical microspheres without apparent morphological changes. Representatively, Ag/AgCl/BiOCH-3 displayed a 3D hierarchical microsphere shape (Figure 2c). Close inspection of the SEM image (Figure 2d) reveals that numerous Ag/AgCl NPs were compactly coated on the 2D nanosheets of BiOCOOH, proving the formation of the Ag/AgCl/BiOCOOH heterojunction.

TEM images were further collected to provide insights into the microstructure of Ag/AgCl/BiOCH-3. As expected, the TEM images (Figure 3a,b) reveal similar results to the SEM images (Figure 2c,d), confirming that the Ag/AgCl NPs (size: ~20 to 70 nm) were closely embedded in a flower-like BiOCOOH via an in-situ growth process. The in-situ deposition of Ag/AgCl NPs on BiOCOOH spheres ensured the compact contact of interfaces between them, benefiting the separation of photo-induced charge carriers and further enhancing the photocatalytic capability of these ternary heterojunctions. Based on these characterizations, it can be inferred that the Ag/AgCl/BiOCOOH ternary heterojunction with tight interfacial contact has been successfully fabricated by a simple in-situ precipitation route.

### 3.2. Optical Properties

The optical responses of BiOCOOH, AgCl, and Ag/AgCl/BiOCOOH heterojunctions were measured by UV‒Vis spectrophotometry (Figure 4). Bare BiOCOOH presents an intense absorption in the UV region up to ~370 nm, in line with that in the reported literature [10]. When integrated with Ag/AgCl NPs, all Ag/AgCl/BiOCOOH heterojunctions display a remarkable improvement in VL absorption, mainly due to the SPR effect of Ag NPs. This fact indicates that Ag/AgCl/BiOCOOH heterojunctions are capable of harvesting more sunlight to efficiently decompose toxic contaminants as compared to pure BiOCOOH. In addition, the absorption edge (λ_g_) of AgCl is ~387 nm. According to the following formula: Eg = 1240/λ_g_, the band gap energy (Eg) values of BiOCOOH and AgCl are determined as 3.40 [10] and 3.21 eV [23], respectively.

### 3.3. Photocatalytic Activity

The photocatalytic capabilities of BiOCOOH, Ag/AgCl, and Ag/AgCl/BiOCOOH heterojunctions were measured for the elimination of RhB. Figure 5a displays the photodegradation rates of RhB over various samples. The blank experiment verified the structural stability of the RhB compound in the absence of any photocatalysts, and the photolysis of RhB was negligible after 60 min of simulated solar irradiation. The photocatalytic activities of the samples followed the order: Ag/AgCl/BiOCH-3 > Ag/AgCl/BiOCH-4 > Ag/AgCl/BiOCH-2 > Ag/AgCl/BiOCH-1 > Ag/AgCl > BiOCOOH > a mixture of Ag/AgCl and BiOCOOH. These Ag/AgCl/BiOCOOH heterojunctions exhibited prominently improved photocatalytic activity than Ag/AgCl and BiOCOOH owing to the formation of the ternary heterostructure. Moreover, the photocatalytic results also signified the significance of the optimum amount of Ag/AgCl toward the photocatalytic activities of the as-prepared heterojunctions. The photocatalytic activity first rose with the increase in Ag/AgCl content. When the Ag/AgCl content reached 20 wt.%, the as-prepared Ag/AgCl/BiOCH-3 was demonstrated to be the best candidate among all the as-prepared catalysts, and it only took about 40 min to completely decompose RhB, which could be credited to the improved photo-absorption capacity (Figure 4) and the markedly suppressed recombination of charge carriers. However, when the content of Ag/AgCl was 30 wt.%, the photocatalytic activity slumped, which could be due to the fact that the aggregation of Ag/AgCl undermined the efficient separation of carriers. Furthermore, a blend sample denoted as a mixture was also applied to degrade the RhB solution, and approximately 67.7% of RhB was eliminated, verifying the establishment of the heterojunction in Ag/AgCl/BiOCH-3 and the eminent role of the heterojunction in determining the photocatalytic performance.

Additionally, according to the formula of −ln (*C*/*C*_0_) = *kt*, the apparent degradation rate constants (*k*) over various catalysts were calculated in Figure 5b. Remarkably, Ag/AgCl/BiOCH-3 bore the highest *k* value of 0.1353 min^−1^ that was around 15.5 or 11.2 times greater than that of pure BiOCOOH (0.0082 min^−1^) or Ag/AgCl (0.0111 min^−1^), respectively.

CIP, a colorless antibiotic, was selected as a probe to further appraise the photocatalytic capability of Ag/AgCl/BiOCH-3 (Figure 6). As expected, the CIP degradation efficiency over Ag/AgCl/BiOCH-3 (86.9%) was much greater than that over BiOCOOH (29.8%), or Ag/AgCl (38.8%), further verifying the high photocatalytic capability of Ag/AgCl/BiOCH-3.

For elucidation of the mineralization capacity of the as-prepared samples, TOC measurements were implemented during the degradation of RhB (10 mg/L, 80 mL) using Ag/AgCl/BiOCH-3, BiOCOOH, or Ag/AgCl as the photocatalyst. As shown in Figure 7a, Ag/AgCl/BiOCH-3 presented an excellent mineralization capability, and about 54.9% of RhB was mineralized after 60 min, which was much higher than that by BiOCOOH (16.7%) or Ag/AgCl (21.6%).

Recyclability of a photocatalyst is another important parameter for practical utilization in industry [29,30]. Hence, recycling experiments for RhB degradation by Ag/AgCl/BiOCH-3 were executed. In each run, 30 min of dark adsorption and 30 min of visible-light irradiation were involved. As displayed in Figure 7b, Ag/AgCl/BiOCH-3 showed satisfactory photocatalytic behavior without apparent loss of activity even after six successive runs. Moreover, the XRD technique was employed to characterize the recycled Ag/AgCl/BiOCH-3 after six runs. As observed in Appendix A, no additional peaks were detected compared to those in the XRD pattern of the fresh Ag/AgCl/BiOCH-3, proving the stable crystalline structure. In other words, the Ag/AgCl/BiOCH-3 ternary heterojunction is a kind of stable photocatalyst that can be utilized for at least six runs with steady photocatalytic activity.

### 3.4. Reaction Mechanism

To elucidate the photocatalytic mechanism of pollutant degradation over the Ag/AgCl/BiOCH-3 ternary heterojunction, radical quenching tests were first conducted by using 1 mM of ammonium oxalate (AO), isopropyl alcohol (IPA), and benzoquinone (BQ) to scavenge h^+^, •OH, and •O_2_^−^ species, respectively (Figure 8) [31]. Clearly, the introduction of AO, IPA, and BQ made the RhB degradation efficiencies drastically decline from 100% to 53.8%, 33.2%, and 58.6%, respectively, verifying that h^+^, •OH, and •O_2_^−^ reactive species were produced and collectively involved in the pollutant degradation by Ag/AgCl/BiOCH-3.

To further illustrate the principle behind the enhanced photocatalytic property of the Ag/AgCl/BiOCH-3 heterojunction, photoluminescence (PL) spectra were measured to analyze the migration and separation of photo-induced carriers [32,33]. In general, the stronger the PL intensity, the faster the recombination speed of the charge carriers, and, hence, the weaker the photocatalytic capability [14,15,29,30,32,33,34]. As displayed in Figure 9, BiOCOOH showed a tough PL peak situated at ~370 nm, indicating that the reunion speed of the electrons and holes was fast. By contrast, Ag/AgCl/BiOCH-3 presented a much weaker emission peak than BiOCOOH, which signified that the reunion of photo-excited carriers was substantially hindered by the introduction of Ag/AgCl NPs, implying the enhanced photocatalytic capability.

It is well known that the band positions of the constituents of the heterojunction play a crucial role in determining the transfer pathway of photogenerated electrons and holes [35]. As reported, the CB potential (*E*_CB_) and VB potential (*E*_VB_) of BiOCOOH are −0.67 and +2.73 V (*versus* Normal Hydrogen Electrode (NHE)), respectively [36], while those of AgCl are −0.06 and +3.15 V (*versus* NHE), respectively [37].

In view of the above characterization, photocatalytic performances, and radical trapping experiment, a plausible photocatalytic mechanism for the elimination of toxic contaminants over the Ag/AgCl/BiOCOOH ternary heterojunction photocatalyst is put forward in Figure 10. The excellent photocatalytic performance of Ag/AgCl/BiOCOOH mainly gives credit to the fascinating ternary heterostructure, which remarkably promotes the separation of charge carriers and ameliorates the sunlight absorption capability [38,39]. With simulated sunlight illumination, both BiOCOOH and AgCl can harvest the photons with energies greater than 3.4 and 3.21 eV to generate electrons and holes on their CB and VB, respectively. The photo-excited electrons on the CB of BiOCOOH can be easily injected into the CB of AgCl. Afterward, the metallic Ag NPs function as electron sinks to accept electrons from the CB of AgCl and, consequently, these accumulated electrons on the surface of Ag NPs can be scavenged by O_2_ to produce •O_2_^−^ reactive species (Figure 8). Notably, the electron transfer process could effectively inhibit the photo-corrosion of AgCl, further contributing to the high stability of the ternary hetero-structure. Meanwhile, the holes on the VB of AgCl tend to thermodynamically flow into the VB of BiOCOOH. Subsequently, parts of the holes left on the VB of BiOCOOH can react with H_2_O to form •OH radicals, while the rest of the holes with a powerful oxidation capacity can directly decompose the pollutants (Figure 8). To sum up, by virtue of the well-matched band alignment of the ternary hetero-structure, the produced h^+^, •OH, and •O_2_^−^ reactive species primarily contributed to the efficient degradation and mineralization of toxic contaminants (RhB/CIP). 

## 4. Conclusions

A novel 3D flower-like Ag/AgCl/BiOCOOH ternary heterojunction photocatalyst was successfully constructed via loading AgCl NPs on BiOCOOH microspheres, followed by photoreduction. The photocatalytic capabilities of these samples were assessed through the photocatalytic destruction of RhB and CIP under simulated sunlight. The Ag/AgCl/BiOCH-3 ternary heterojunction photocatalyst displayed the best photocatalytic capability, and the RhB and CIP degradation efficiencies could achieve 100% and 86.9%, respectively. The extraordinary photocatalytic capability was primarily credited to the merits of the enhanced sunlight harvesting capability and evidently suppressed recombination of photo-induced carriers, originating from the synergistic effect of the novel ternary heterojunction system. The trapping experiments demonstrated that h^+^, •OH, and •O_2_^−^ reactive species collaboratively contributed to the degradation of pollutants. Moreover, Ag/AgCl/BiOCH-3 possessed a good stability and strong mineralization capability. Therefore, this work not only offers an excellent ternary heterojunction photocatalyst, but might also provide a facile method for constructing 3D microsphere ternary heterojunction catalysts for photocatalytic applications.

## Figures and Tables

**Figure 1 nanomaterials-09-01562-f001:**
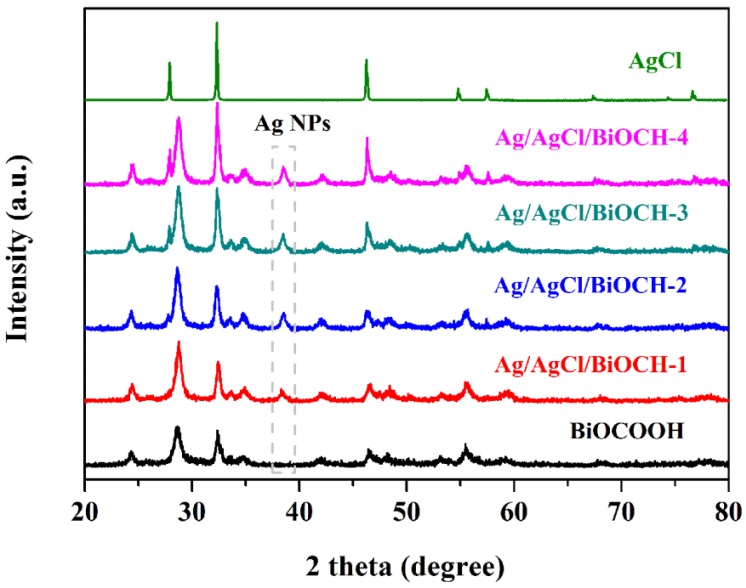
XRD patterns of BiOCOOH, AgCl, and Ag/AgCl/BiOCOOH heterojunctions (Ag/AgCl/BiOCH-1, Ag/AgCl/BiOCH-2, Ag/AgCl/BiOCH-3, and Ag/AgCl/BiOCH-4).

**Figure 2 nanomaterials-09-01562-f002:**
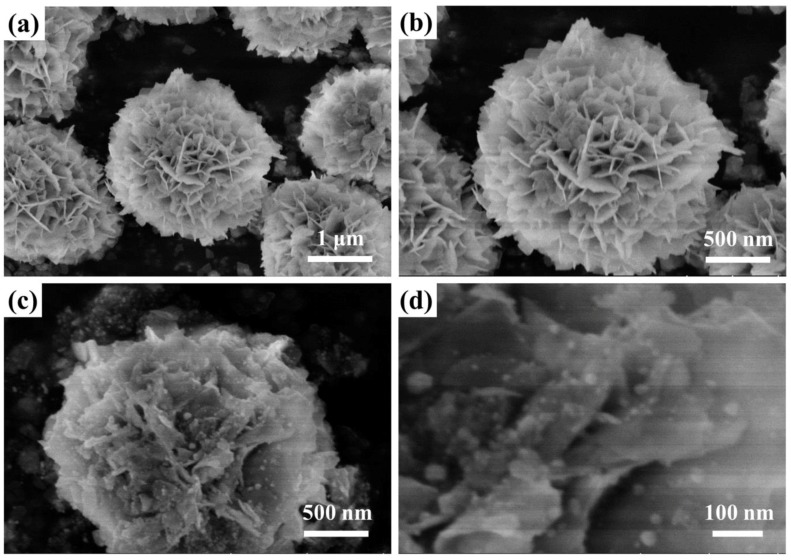
SEM images of (**a**,**b**) BiOCOOH and (**c**,**d**) Ag/AgCl/BiOCH-3.

**Figure 3 nanomaterials-09-01562-f003:**
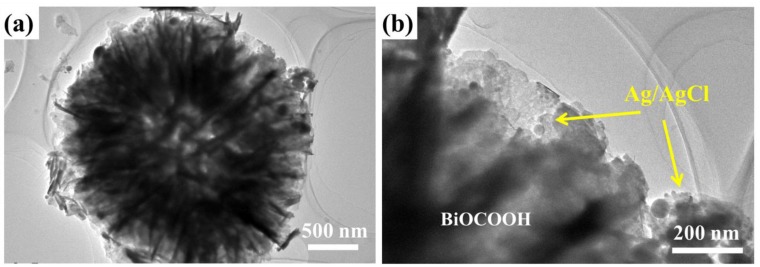
(**a**,**b**) TEM images of Ag/AgCl/BiOCH-3.

**Figure 4 nanomaterials-09-01562-f004:**
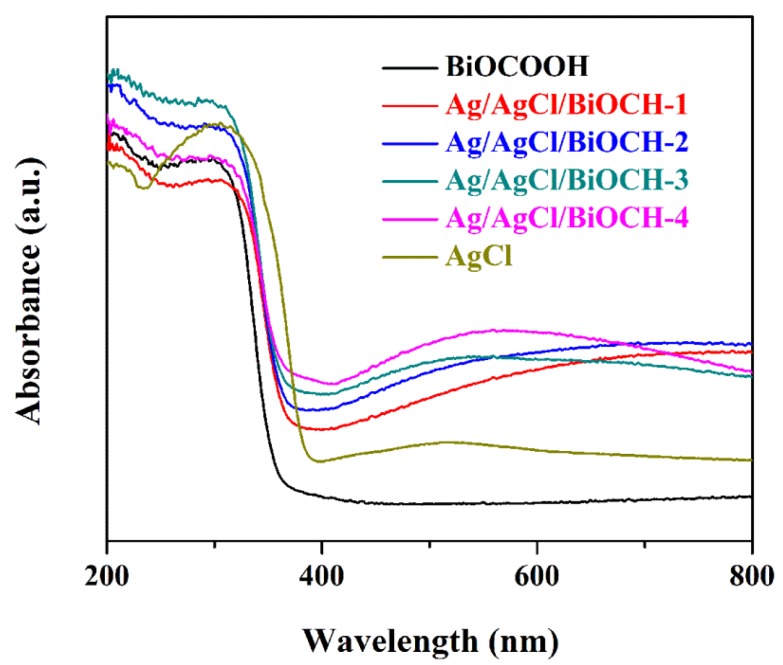
UV‒vis DRS of BiOCOOH, AgCl, and Ag/AgCl/BiOCOOH heterojunctions (Ag/AgCl/BiOCH-1, Ag/AgCl/BiOCH-2, Ag/AgCl/BiOCH-3, and Ag/AgCl/BiOCH-4).

**Figure 5 nanomaterials-09-01562-f005:**
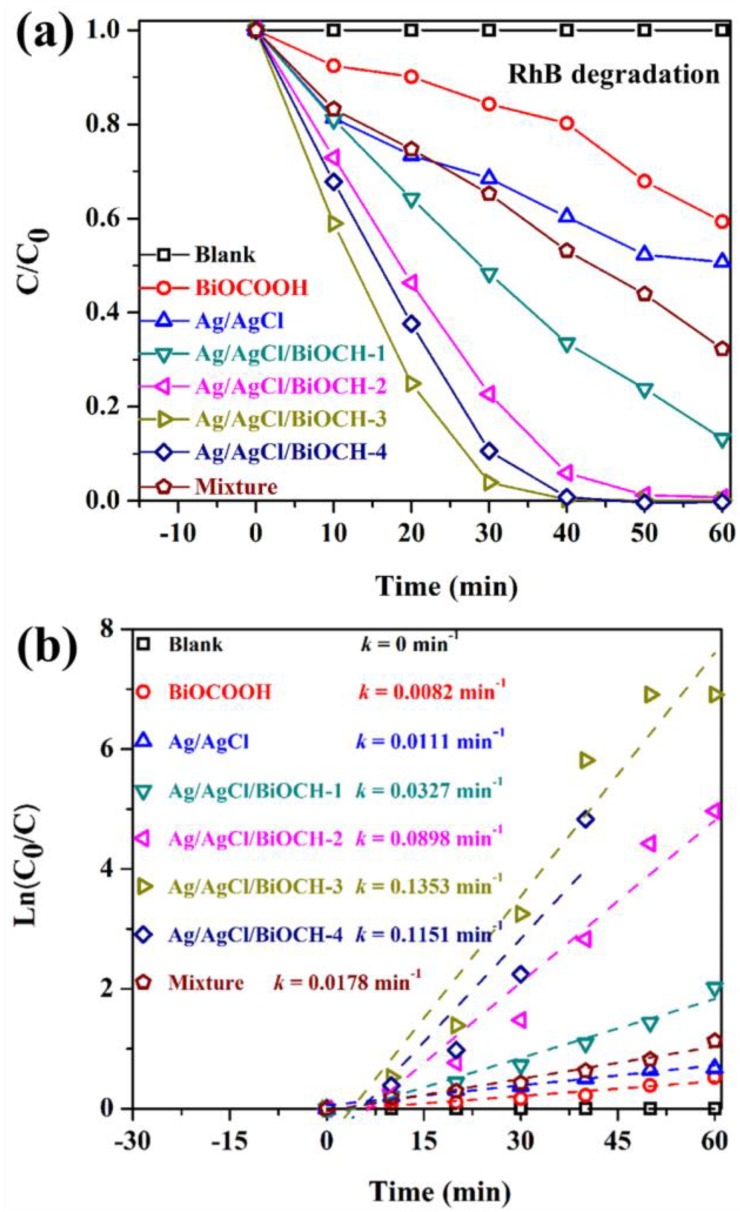
(**a**,**b**) Photocatalytic removal of rhodamine B (RhB) (10 mg/L, 80 mL) over as-obtained photocatalysts (30 mg) under simulated sunlight.

**Figure 6 nanomaterials-09-01562-f006:**
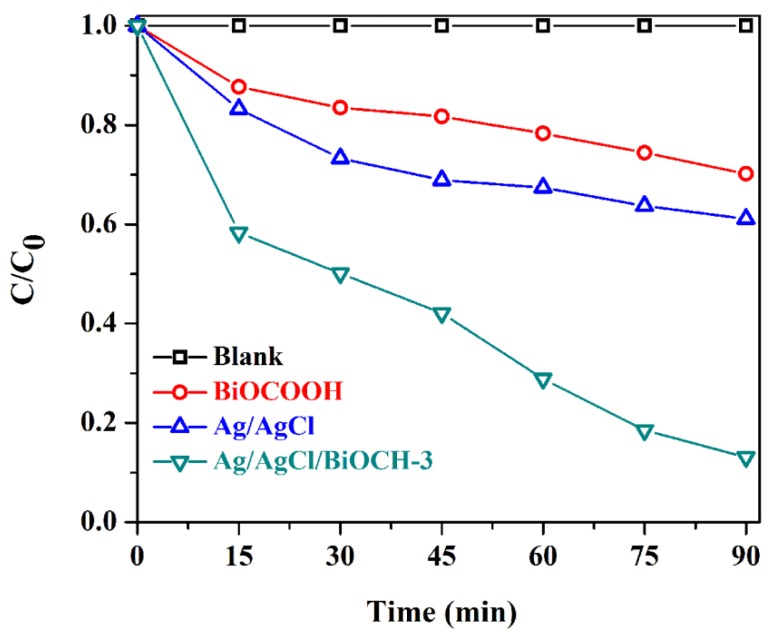
Photocatalytic removal of ciprofloxacin (CIP) (10 mg/L, 80 mL) over different photocatalysts (30 mg) under simulated sunlight.

**Figure 7 nanomaterials-09-01562-f007:**
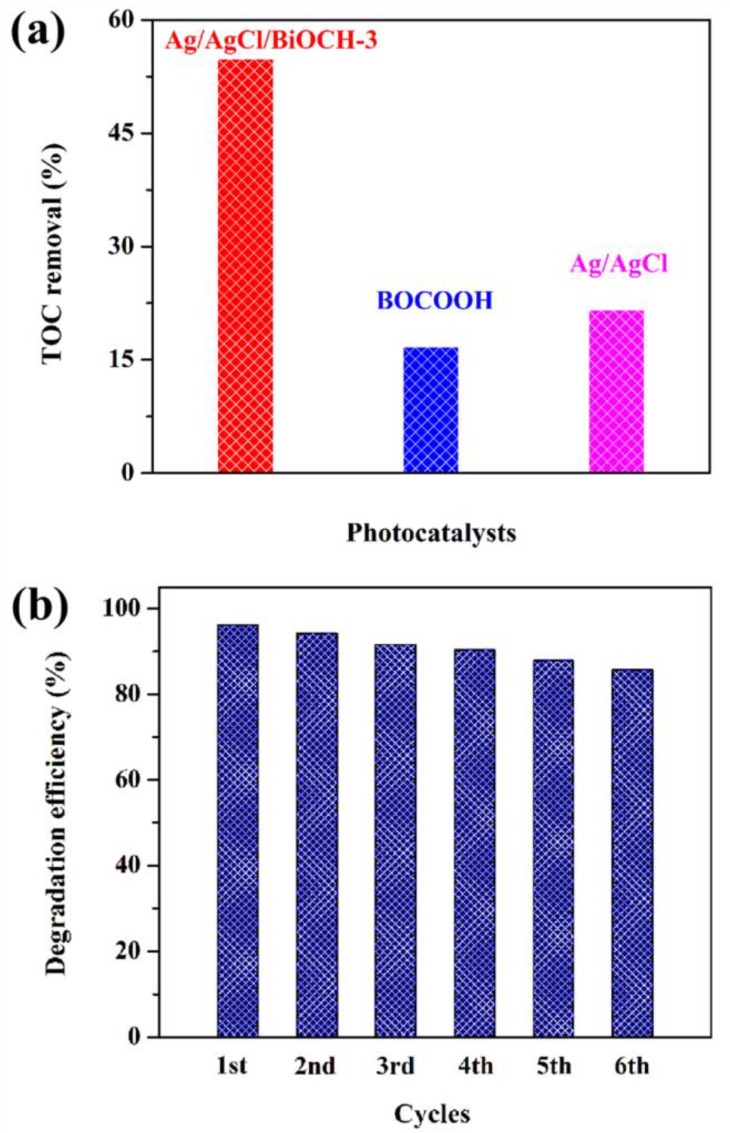
(**a**) TOC removal of RhB (10 mg/L, 80 mL) over Ag/AgCl/BiOCH-3 (30 mg), BiOCOOH (30 mg), or Ag/AgCl (30 mg) after 60 min of illumination; (**b**) recycling tests of Ag/AgCl/BiOCH-3 (30 mg) for RhB (10 mg/L, 80 mL) removal.

**Figure 8 nanomaterials-09-01562-f008:**
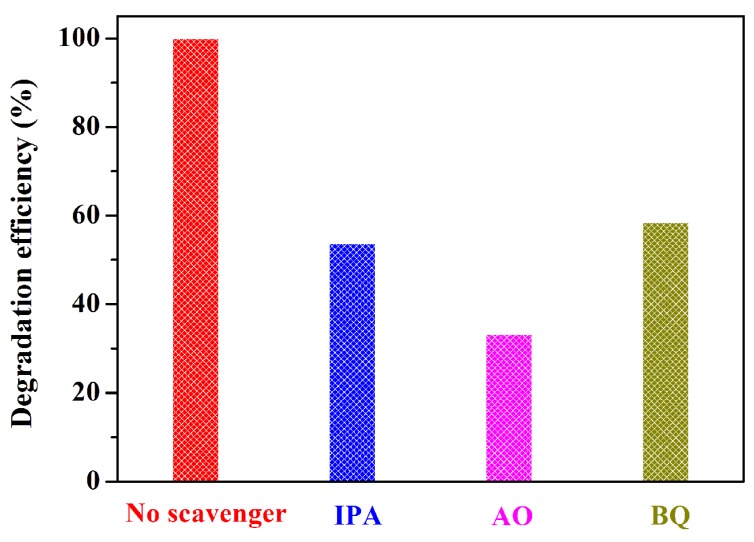
Influences of different quenchers on the photocatalytic capability of Ag/AgCl/BiOCH-3.

**Figure 9 nanomaterials-09-01562-f009:**
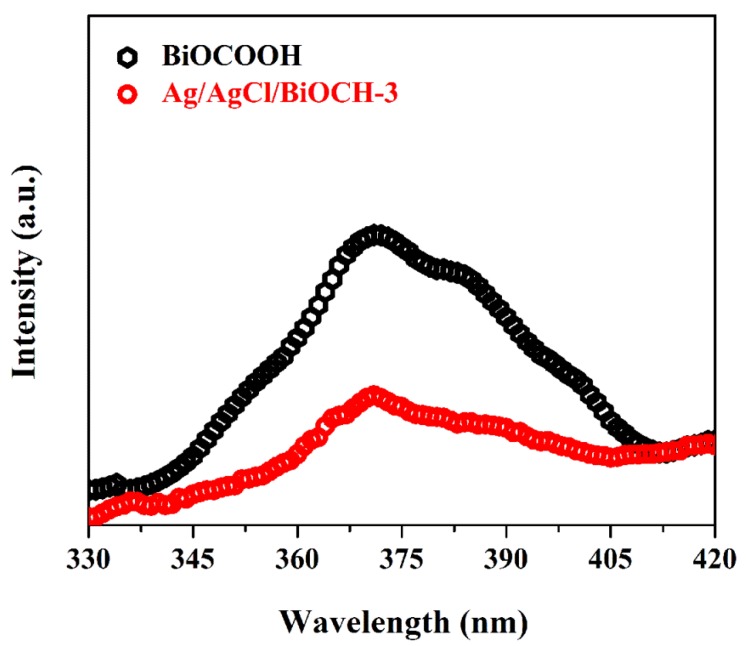
PL spectra of BiOCOOH and Ag/AgCl/BiOCH-3.

**Figure 10 nanomaterials-09-01562-f010:**
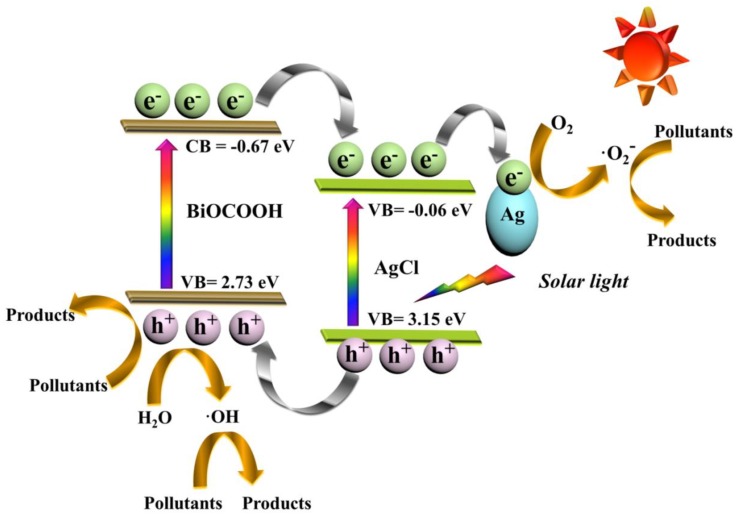
Photocatalytic mechanism of the ternary Ag/AgCl/BiOCOOH heterojunction photocatalyst under simulated sunlight.

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
