# Peer review of "A Novel Flower-Like Ag/AgCl/BiOCOOH Ternary Heterojunction Photocatalyst: Facile Construction and Its Superior Photocatalytic Performance for the Removal of Toxic Pollutants"

_nanomaterials, 2019, doi:10.3390/nano9111562_

Round 1

Reviewer 1 Report

The authors reported flower-like Ag/AgCl/BiOCOOH ternary heterojunction photocatalysts and demonstrated their photocatalytic activity for removal of pollutants, rhodamine B and ciprofloxacin.

The submitted work represents an interesting and complex study, however there are certain issues that authors should address prior further publication.

Lines 91-95: In the lines 92-93, the Authors stated the 30 wt% Ag/AgCl/BiOCOOH heterojunction was denoted as Ag/AgCl/BiOCH-3, however within the lines 94-95 the same name is given for the 20 wt% Ag/AgCl/BiOCOOH heterojunction. Therefore, within the discussion part one is wondering which catalysts exactly have been used in certain points and thus the reader may be confused. Figure 5 (b) The points plotted in the Fig.5(b) should be exactly within the range of ln(C0/C) vs. time where they occur. There is no need to show the range up to 20 of ln(C0/C); up to 8 would be just enough. Thus one could better compare the differences between each of the examined heterojunction. While authors describe utilization of the series of scavengers for elucidation of the photocatalytic mechanism of pollutants, it seems that there is a missing reference for other examples from the literature, where such scavengers were used for that purpose. The following reference might be cited: Water, 2019, 11(5), 953; http://doi.org/10.3390/w11050953. There is a missing reference when the empirical formula for estimation of the valence and conduction bands is given. Furthermore, it is not clear whether authors estimated those values by themselves from the electrochemical studies or the given values were based on the literature. Therefore, in the section 2 no information on the electrochemical measurements were described. Please specify that in the paper.

The described issues that authors are expected to address do not change the fact that authors have presented interesting and valuable studies, that are worth to be published in the Nanomaterials after minor revision.

Author Response

Lines 91-95: In the lines 92-93, the Authors stated the 30 wt% Ag/AgCl/BiOCOOH heterojunction was denoted as Ag/AgCl/BiOCH-3, however within the lines 94-95 the same name is given for the 20 wt% Ag/AgCl/BiOCOOH heterojunction. Therefore, within the discussion part one is wondering which catalysts exactly have been used in certain points and thus the reader may be confused.

Answer: Thank you for your kind suggestions. In the revised paper, I have updated it as follows:

“The as-prepared 30wt% Ag/AgCl/BiOCOOH was washed with deionized water thoroughly and dried at 70 °C in a vacuum oven and denoted as Ag/AgCl/BiOCH-4. Similarly, 5wt% Ag/AgCl/BiOCOOH (Ag/AgCl/BiOCH-1), 10 wt% Ag/AgCl/BiOCOOH (Ag/AgCl/BiOCH-2), 20 wt% Ag/AgCl/BiOCOOH (Ag/AgCl/BiOCH-3), AgCl and Ag/AgCl were also synthesized.”

Figure 5 (b) The points plotted in the Fig.5(b) should be exactly within the range of ln(C0/C) vs. time where they occur. There is no need to show the range up to 20 of ln(C0/C); up to 8 would be just enough. Thus one could better compare the differences between each of the examined heterojunction.

Answer: Thank you for your professional suggestions. We have updated the Figure 5b.

Figure 5.(a, b) Photocatalytic removal of RhB (10 mg/L, 80 mL) over as-obtained photocatalysts (30 mg) under simulated sunlight.

While authors describe utilization of the series of scavengers for elucidation of the photocatalytic mechanism of pollutants, it seems that there is a missing reference for other examples from the literature, where such scavengers were used for that purpose. The following reference might be cited: Water, 2019, 11(5), 953; http://doi.org/10.3390/w11050953.

Answer: Thank you for your professional suggestions. We have cited the reference.

Regulska, E.; Breczko, J.; Basa, A., Pristine and graphene-quantum-dots-decorated spinel nickel aluminate for water remediation from dyes and toxic pollutants Water 2019, 11, 953.

There is a missing reference when the empirical formula for estimation of the valence and conduction bands is given. Furthermore, it is not clear whether authors estimated those values by themselves from the electrochemical studies or the given values were based on the literature. Therefore, in the section 2 no information on the electrochemical measurements were described. Please specify that in the paper.

Answer: Thank you for your professional suggestions.

The described issues that authors are expected to address do not change the fact that authors have presented interesting and valuable studies, that are worth to be published in the Nanomaterials after minor revision.

Answer: Thank you for your support.

Reviewer 2 Report

The authors prepared flower-like Ag/AgCl/BiOCOOH ternary
heterojunction and investigated the elimination of RhB and ciprofloxacin (CIP) under simulated sunlight . The prepared photoactalyst were well characterized.However, the paper it is still insufficient in the listed aspects:

There is lack of emission spectra of  simulated
 sunlight source. Is it not known if a light filter should be used to remove  UV light? This information is important for the readers because without this information it is not certain that the purpose of these studies was achieved. In my opinion the novelty of obtained photocatalysts or methods should be strongly highlighted in introduction.  Have you performed other experiments in order to demonstrate that obtained catalyst have good stability and are environmentally friendly? I am referring to release of Ag to the environment can causes toxicity problems. Results and discussion: - There is no discussion in the chapter at all, comparison with existing literature is missing, limitations of the results should be discussed, future research should be outlined.

Author Response

Answer: Thank you for your professional suggestions. Actually, the simulated UV and visible light was used as the light source, so a 300 W Xe lamp (NeT HSX-300, Beijing, China) without a light filter to remove the UV light was employed as the simulated sunlight source.

Furthermore, we have updated the introduction to highlight the novelty of the samples.

Up to now, Ag/AgCl has been widely utilized to modify different semiconductors for triggering extraordinary photocatalytic capability because of its ability to efficaciously improve the visible light (VL) response by the surface plasmon resonance (SPR) effect of metallic Ag and boost the separation of electrons and holes [19-24]. For instance, the Ag/AgCl/titanium phosphate [25], Ag/AgCl/TiO2 [22] and Ag/AgCl/SrTiO3 [21]  have been constructed and displayed superior VL photocatalytic activity. Based on the band theory, the band structures of Ag/AgCl and BiOCOOH are suitable for the generation of a novel plasmonic “type II” heterojunction, which could efficiently promote the separation of photo-excited charge carriers during the photocatalytic process. Enlighten by the above consideration, the novel hierarchical heterostructure of BiOCOOH decorated with Ag/AgCl nanoparticles can be expected to be a highly efficient photocatalyst for pollutant elimination under simulated sunlight.

Herein, in this work, a novel 3D flowerlike Ag/AgCl/BiOCOOH ternary heterojunction photocatalyst has been successfully synthesized through a facile solvothermal-precipitation-photo-reduction route. Ag/AgCl NPs were evenly deposited BiOCOOH micro-flowers, enabling fast separation of carriers and better absorption of light via multi-reflection of light. The underlying properties and photocatalytic capability toward removal of rhodamine B (RhB) and ciprofloxacin (CIP) in aqueous solution were systematically investigated. The possible photocatalysis mechanism regarding the transfer and segregation of charge carriers was proposed. This study offers a deep insight into the design of highly efficient ternary heterojunction photocatalysts for wastewater treatment.

Due to the limited resources, we just test the photocatalytic stability of the samples.

Recyclability of a photocatalyst is another premier parameter for practical utilizations in industries [29, 30]. Hence, the recycling experiments for RhB degradation by Ag/AgCl/BiOCH-3 were executed and the results were shown in Figure 7b. The photocatalytic activity maintained well up to the forth run and then slightly descended in the last two runs, demonstrating the good stability (Figure 7b). Moreover, the XRD technique was employed to characterize the recycled Ag/AgCl/BiOCH-3 after six runs. As observed in Figure S1, no additional peaks were detected compared with the ones in the XRD pattern of the fresh Ag/AgCl/BiOCH-3, evidencing the stable crystalline structure. In a word, the Ag/AgCl/BiOCH-3 ternary heterojunction is a kind of stable photocatalysts that can be utilized for at least six runs with steady photocatalytic activity. Moreover, the electron transfer process could effectively inhibit the photo-corrosion of AgCl, further contributing to the high stability of the ternary hetero-structure.

Reviewer 3 Report

The manuscript deals with synthesis, characterization and photocatalytic testing of the ternary heterojunction photocatalyst Ag/AgCl/BiOCOOH. The presented results show that photocatalytic performance of the prepared materials depends on the amount of Ag/AgCl supported on the BiOCOOH surface. Moreover, from results of quenching experiments and band position calculation a photocatalytic mechanism was suggested. The presented results are interesting; however, the manuscript shows a lack in experimental description.

Page 3 line 111

The authors wrote that the concentration of RhB and CIP in the solution was measured with UV-vis spectrometer. This has to be explained in more detail. After irradiation with simulated sunlight the liquid phase contains beside RhB or CIP also their degradation products which can be colored too. How was the concentration of RhB or CIP extracted from the UV-vis spectrum showing a superposition in UV-vis absorption of different compounds?

Page 4 Figure 1

It is possible to roughly determine the size of the silver nanoparticles in the different samples from the silver reflection in the XRD powder pattern using the Scherrer equation?

Page 8 Figure 5

How was the value c/c0 calculated? This should be explained in the experimental section.

Page 9 Figure 6

How was the value c/c0 calculated? This should be explained in the experimental section.

Page 10 Figure 7

From the results presented in Figure 7 it cannot be excluded that catalyst activity decreased already after the first run because in the first four experiments degradation efficiency was always 100 % and only one point was presented. The authors should repeat this experiment and start with degradation efficiency of 90 %. The experimental conditions how the recycling test was performed have to be also given? How was the catalyst separated from the solution? Was the catalyst washed between the single runs? How long was the suspension irradiated?

Page 11 line 235

It was not reported in the experimental section how the quencher experiments were performed. What was the quencher concentration? How long was the sample irradiated? How was the degradation efficiency calculated?

Page 12 line 256

Empirical formulas for calculation of the CB and VB potentials are presented. However, the meaning of the single parameters X, E0 and Eg was not explained. Moreover, the X, E0 and Eg values to calculate CB and VB potentials of BiOCOOH and AgCl were not given. Were the values taken from literature or have the authors carried out Mott-Schottky plots or cyclic voltammetry measurements.

Author Response

The authors wrote that the concentration of RhB and CIP in the solution was measured with UV-vis spectrometer. This has to be explained in more detail. After irradiation with simulated sunlight the liquid phase contains beside RhB or CIP also their degradation products which can be colored too. How was the concentration of RhB or CIP extracted from the UV-vis spectrum showing a superposition in UV-vis absorption of different compounds?

Answer: Thank you for your professional suggestions. In the revised manuscript, this part has been updated as follows: “During the photocatalytic reaction, 3 mL of suspension was collected at a predetermined time interval and centrifuged (5000 rpm, 5 min) to separate aliquot from solid particles for further assessment of the photodegradation rate. The absorbance was monitored by using an UV-vis spectrophotometer (Shimadzu UV-2600) and the normalized concentration variations of RhB and CIP were obtained based on their corresponding maximum absorption at 554 nm and 276 nm.” This method has been widely used in reported references (Applied Surface Science 413 (2017) 372–380ï¼›Applied Surface Science 437 (2018) 51–61; Chem. Eng. J. 358 (2019) 891-902; Journal of Hazardous Materials 364 (2019) 691-699).

Page 4 Figure 1

It is possible to roughly determine the size of the silver nanoparticles in the different samples from the silver reflection in the XRD powder pattern using the Scherrer equation?

Answer: Thank you for your professional suggestions. Actually, due to the weak intensity of Ag peak, it is difficult to determine the size of silver nanoparticles. TEM images were further collected and revealed that the size of Ag/AgCl NPs were about ~20-70 nm, which were closely embedded in flower-like BiOCOOH via an in-situ growth process.

Page 8 Figure 5

How was the value c/c0 calculated? This should be explained in the experimental section.

Answer: Thank you for your professional suggestions. In the revised manuscript, this part has been updated as follows: “The degradation rate (η) of RhB or CIP was estimated by the equation: η = (C0 – Ct)∕C0 × 100% where C0 and Ct were separately the initial concentration of the pollutant and the concentration of pollutant solution at time t.

Page 9 Figure 6

How was the value c/c0 calculated? This should be explained in the experimental section.

Answer: Thank you for your professional suggestions. In the revised manuscript, this part has been updated as follows: “The degradation rate (η) of RhB or CIP was estimated by the equation: η = (C0 – Ct)∕C0 × 100% where C0 and Ct were separately the initial concentration of the pollutant and the concentration of pollutant solution at time t.”

Page 10 Figure 7

From the results presented in Figure 7 it cannot be excluded that catalyst activity decreased already after the first run because in the first four experiments degradation efficiency was always 100 % and only one point was presented. The authors should repeat this experiment and start with degradation efficiency of 90 %. The experimental conditions how the recycling test was performed have to be also given? How was the catalyst separated from the solution? Was the catalyst washed between the single runs? How long was the suspension irradiated?

Answer: Thank you for your professional suggestions. In the revised manuscript, this part has been updated as follows: “

The reusability and stability of Ag/AgCl/BiOCH-3 was measured via repeating the uniform experiment for six runs, and each run involved 30 min of dark adsorption and 30 min of reaction. After each run, the photocatalyst was first separated from RhB aqueous solution via centrifugation (5000 rpm, 3 min), and then rinsed thoroughly with deionized water and ethanol, dried at 70 °C for 10 h, and subjected to the subsequent run.

Hence, the recycling experiments for RhB degradation by Ag/AgCl/BiOCH-3 were executed. In each run, 30 min of dark adsorption and 30 min of visible-light irradiation were involved. As displayed in Figure 7b, Ag/AgCl/BiOCH-3 showed satisfactory photocatalytic behavior without apparent loss of activity even after six successive runs. Moreover, the XRD technique was employed to characterize the recycled Ag/AgCl/BiOCH-3 after six runs. As observed in Figure S1, no additional peaks were detected compared with the ones in the XRD pattern of the fresh Ag/AgCl/BiOCH-3, evidencing the stable crystalline structure. In a word, the Ag/AgCl/BiOCH-3 ternary heterojunction is a kind of stable photocatalysts that can be utilized for at least six runs with steady photocatalytic activity.

Page 11 line 235

It was not reported in the experimental section how the quencher experiments were performed. What was the quencher concentration? How long was the sample irradiated? How was the degradation efficiency calculated?

Answer: Radical-trapping experiments were executed via introducing the scavenger of ammonium oxalate (AO, 1 mmol, quencher of h+), benzoquinone (BQ, 0.02 mmol, quencher of •O2), or isopropyl alcohol (IPA, 1 mmol, quencher of •OH) into the photocatalytic system.

Each experiment involved 30 min of dark adsorption and 60 min of visible-light reaction.

 “The degradation efficiency (η) of RhB was estimated by the equation: η = (C0 – Ct)∕C0 × 100% where C0 and Ct  are separately the absorbance of the RhB solution before irradiation and the absorbance of the RhB solution at irradiation time t.”

Page 12 line 256

Empirical formulas for calculation of the CB and VB potentials are presented. However, the meaning of the single parameters X, E0 and Eg was not explained. Moreover, the X, E0 and Eg values to calculate CB and VB potentials of BiOCOOH and AgCl were not given. Were the values taken from literature or have the authors carried out Mott-Schottky plots or cyclic voltammetry measurements.

Answer: Thank you for your professional suggestions. In the revised manuscript, this part has been updated as follows: “

“It is well known that the band positions of the constituents of the heterojunction play a crucial role in determining the transfer pathway of photo-generated electrons and holes [35]. As reported, the CB potential (ECB) and VB potential (EVB) of BiOCOOH are determined as -0.67 V and +2.73 V (versus NHE), respectively [36], while those of AgCl are -0.06 V and +3.15 V (versus NHE), respectively [37].”

Round 2

Reviewer 3 Report

All my questions and comments were answered. The manuscript can be published.